# The Impact of Oral Health on the Number of Missing Matches and Physical Performance in Elite Male Soccer Players

**DOI:** 10.3390/sports13120417

**Published:** 2025-12-01

**Authors:** René Schwesig, Stephan Schulze, Lars Reinhardt, Eduard Kurz, Andreas Wienke, Thomas Bartels, John Brandes, Christian Ralf Gernhardt

**Affiliations:** 1Department of Orthopaedic and Trauma Surgery, Martin-Luther-University Halle-Wittenberg, University Medicine Halle, 06120 Halle, Germany; rene.schwesig@uk-halle.de (R.S.); stephan.schulze@uk-halle.de (S.S.); lars.reinhardt@uk-halle.de (L.R.); eduard.kurz@uk-halle.de (E.K.); 2Department for Diagnostic and Interventional Radiology, University Hospital Halle (Saale), 06120 Halle, Germany; 3Institute of Medical Epidemiology, Biostatistics and Informatics, Martin-Luther-University Halle-Wittenberg, University Medicine Halle, 06112 Halle, Germany; andreas.wienke@uk-halle.de; 4MVZ Sports Clinic Halle GmbH, Center of Joint Surgery, 06108 Halle, Germany; thomas.bartels@sportklinik-halle.de; 5Health Services Research Group, Department of Internal Medicine, Martin-Luther-University Halle-Wittenberg, University Medicine Halle, 06112 Halle, Germany; john.brandes@uk-halle.de; 6University Outpatient Clinic for Conservative Dentistry and Periodontology, Martin-Luther-University Halle-Wittenberg, University Medicine Halle, 06112 Halle, Germany

**Keywords:** soccer, physical performance, oral health status, DMFT, periodontal health, oral hygiene, grip strength, jumping performance, endurance

## Abstract

Background: Oral health (OH) seems to be relevant for the number of sick and injured days (NSID), missing matches (MM) and secondarily for the physical performance (PP). Aim: The objective was to clarify possible associations between OH, NSID, and PP for elite soccer players. Methods: Thirty-nine male athletes (age: 24.6 ± 4.2 years, age range: 17–34 years) from a third league professional soccer team were examined concerning several dental parameters (decayed, missing, and filled teeth, DMFT; periodontal screening index, PSI; approximal plaque index, API; papillary bleeding index, PBI) and PP parameters. The PP diagnostic contains grip strength, posturography, jump and sprint tests, and an endurance test on treadmill. Furthermore, the number of sick and injured days and missing matches was collected and assessed over four seasons. Results: We could not find any relevant (r > 0.7) correlations between sick and injured days or missing matches and variables of different dimensions (OH, PP). The soccer players showed a remarkable level of oral health (missing teeth, MT: 0.18 ± 0.56), jumping performance (44.5 ± 5.42 cm), and grip strength (53.7 ± 7.02 kg). The endurance capacity (velocity at 4 mmol/L lactate threshold, v4: 14.9 ± 1.11 km/h) was on an average level, whereas the levels of postural stability (stability indicator, ST: 20.0 ± 4.55) and sprinting performance (10 m sprint: 1.79 ± 0.09 s) were comparatively low. Only five players (13%) reported that oral health had ever had a negative impact on his physical performance. Two players (5%) reported currently tooth pain and six players (15%) bleeding gums or grinding teeth. Conclusions: Based on the high level of dental health, it was difficult to prove any relationships between OH and the NSID/MM or PP. Nevertheless, it seems that young soccer players benefit particularly from improved oral health programs. The excellent dental care appears to have a positive effect on general health and physical performance in soccer.

## 1. Introduction

In high-performance environments, elite soccer players are expected to maintain peak physical, mental, and physiological conditions [1]. However, a growing body of evidence indicates that oral health encompassing dental caries, periodontal disease, erosion, and malocclusion may exert a substantial but under recognized influence on athlete well-being and physical performance [2,3]. The international literature regarding this topic, with a focus on professional soccer and related elite sports, might help to illuminate this critical but often overlooked area [4].

Formerly published clinical research among elite soccer players revealed a high prevalence of oral pathologies including erosions, caries and periodontitis and also functional problems [4]. Needleman et al. [5] examined 187 elite soccer male players from the Premier League and Championship clubs and found 37% had active dental caries, 53% exhibited dental erosion, and 5% suffered moderate-to-severe periodontal disease. Self-reported impairments regarding performance during training or within regular matches exceeded 7% [2]. Similar findings emerged from sports medicine clinics: Needleman et al. [5] reported that 302 elite athletes including soccer players experienced substantial rates of caries and periodontal disease, with 6% acknowledging reduced performance due to oral health issues.

Specific investigations into professional soccer players reinforce these observations. A longitudinal study of well-known Spanish soccer team players across three seasons found associations between poor oral hygiene (plaque index, periodontal pockets) and the kinds of re-injury, especially muscle or tendon re-injury [6,7]. In southern Italy, a cohort of 160 professional players exhibited 48% prevalence of dental erosion, 50% periodontal disease, and more than 31% reported oral pain affecting their quality of life and performance [8].

A study of 160 elite youth soccer players in the UK reported that 77% had gingivitis, 23% irreversible periodontitis, and 31% visible decay conditions that impaired their training and competition performance [9]. Recently published meta-analytic results across sports (five studies included soccer players) also supported a performance link [10]. They indicated periodontal disease increased the odds of reduced self-perceived performance by 1.55-fold [10]. Several reviews identified caries and periodontal pathologies as prevalent in elite athletes and connected to systemic inflammation potentially impairing exercise capacity [11,12].

The mechanisms linking poor oral health to physical performance are multifaceted. Chronic dental infections elevate systemic cytokines (e.g., IL-6, TNF α), which contribute to muscle fatigue, oxidative stress, and impaired recovery [13]. The specific conditions of professional athletes including adjusted nutrition, mental stress and increased activity often combined with high loads might also have an impact on the composition and dysbiosis of the oral microbiome resulting in higher risk for dental disease like caries and periodontitis [14,15,16]. Untreated progression of these dental diseases can affect the entire organism, as bacteria can spread from the oral cavity throughout the organism via the bloodstream causing a bacteremia, increasing the levels of circulating inflammatory markers, which might negatively influence physical performance [17,18]. Furthermore, other oral conditions like malocclusion, bruxism, and temporomandibular disorders (TMD) may alter mastication efficiency, cause pain, and influence nutrient absorption—all factors essential for physical resilience especially during the demands of a nearly daily elite training and high physical loads [11,19,20]. Athletes in particular, who often report teeth grinding and bruxism during training and under high physical stress, may have an increased risk of TMD [21]. Recently published results showed that 10% of the investigated 337 athletes reported TMD symptoms [22]. Other publications showed even higher prevalence scores [23]. It is well known that the co-occurrence of bruxism and TMD result in higher prevalence rates, not only in sports but also in an average population [24].

Further results focused on sport specific behavior that exacerbates oral disease. Repeated consumption of acidic carbohydrate-rich sports and energy drinks was linked to enamel erosion and caries in footballers [25]. Reduced salivary flow during intense exercise common among endurance-focused soccer players combined with inadequate sugar intake further elevates susceptibility to tooth decay [26,27,28]. Despite adequate oral hygiene procedures like teeth brushing being common among athletes, many fail to counterbalance dietary and training-related risks [27]. In a Dutch Olympic cohort, over 90% of athletes brushed twice daily, yet more than 80% regularly consumed sports nutrition products linked to caries and erosion [25]. Further large cross-sectional study among elite athletes found oral-health-related quality of life diminished in up to one-third of participants [8,29].

Biological markers support these findings. Studies showed links between periodontal inflammation and elevated C-reactive protein and TNF α levels, factors associated with impaired muscle regeneration and slower recovery [4,30]. Furthermore, dental screening in elite athletes has uncovered associations between gingivitis severity and time lost to injury [31]. In light of these findings, preventive strategies are being explored and demonstrated that implementation of tailored oral health education among professional athletes improved periodontal outcomes and reduced inflammatory biomarkers [32]. Meanwhile, integrating dental care into high-performance programs, such as recently reported in the case of FC Barcelona’s performance diagnostics reflects a growing recognition of the oral-performance connection [4,6,7]. Compared to general studies including not only elite athletes or soccer players the overall global oral health situation is still challenging [33]. In Germany, actual studies showed positive developments regarding oral health conditions in general [34]. However, in spite of proportional reductions, the polarized caries distribution following socio-economic parameters reveals the need for specialized preventive care concepts [35].

Regarding the published results and findings, the present work is based on the current knowledge in scientific sport dentistry and investigated the oral health to physical performance connection among elite soccer players of a third league German team using standardized oral health indices (DMFT, PSI, API, PBI) and different established performance metrics (e.g., injury incidence, number of missing matches, endurance capacity, sprinting and jumping performance).

The primary goal of the present investigation was to evaluate the oral health conditions of professional soccer player in a third league team in Germany. The second aim was to explore possible associations between oral health, physical performance parameters, and the injury incidence.

We hypothesized that oral health had an influence on the number of days on which participation in training sessions was not possible, as well as on the number of matches missed. Secondly, we also assumed that these oral health conditions have a relevant impact on the physical performance of the included athletes.

## 2. Materials and Methods

### 2.1. General Study Design

The comprehensive data collection was planned as a retrospective cohort study over four seasons (Figure 1) in order to realize a sufficient sample size (*n* = 39). Over a period of four years, the number of sick or injured days as well as the number of missed matches was collected using web-based statistical data.

The complex data collection was divided in three parts (Figure 1):Medical and dental anamnesis and clinical examination by a qualified and experienced dentist;Basic physical diagnostic: Anthropometry, grip strength, posturography;Specific performance diagnostic: Sprint and jump tests, endurance test on treadmill including heart rate, and lactate measurement.

Apart from the sprint and jump tests, all these examinations were conducted at the beginning of the preparation phase of each season (last two weeks in June). Sprint diagnostics were performed during the second or third week of the season (mid-August). This should be the peak of physical performance after the preparation phase (six weeks) and the first weeks of the season. All medical and performance diagnostic tests were carried out by examiners trained and familiar with the different tests and procedures. In order to avoid major circadian influences, all mentioned tests were conducted between 9:00 a.m. and 3:00 p.m.

In the last step, the above mentioned publicly visible statistical data was used to record the number of missed days and matches. For this purpose, we used the values from one up to four seasons and then normalized the values to one season. In this way, the comparison of all collected values was possible.

The study protocol was approved by the local Ethics Committee of the Martin Luther University Halle-Wittenberg (reference number: 2022-011, Date of approval: 18 March 2022) and complied with the Declaration of Helsinki [36]. All athletes were teammates of a professional local third league soccer team in Germany.

Before examinations, all players were informed in detail and received written information about the entire project. All participants signed the informed consent statement. Two players were younger than 18 years. Therefore, we provided the written information and the informed consent statement also for their parents or legal guardians. In these two cases, the informed consent statement was signed by their parents. After the signature of the informed consent statement as indicator for their willingness to be part of the investigation, the soccer players were accepted and included in the study group. Moreover, the players were motivated to give maximal effort with strong verbal encouragement during all tests.

### 2.2. Participants

Thirty-nine male elite soccer players (Table 1) were included in this study. All athletes were teammates on a local third-division soccer team, which had been receiving performance diagnostics support for many years as part of a cooperation agreement. This collaboration made it possible to generate a sufficiently large sample over several years. The included soccer players were initially instructed in detail concerning all parts of the investigation. They were trained to perform the tests as accurately as possible. For players under the age of 18 (*n* = 2), the consent of their parents or legal guardians was obtained.

### 2.3. Testing Procedures

#### 2.3.1. Medical and Dental History

Evaluating the oral health status and to become an initial overview of the medical and dental situation, a questionnaire was used to record key points from the medical and dental history of the included professional soccer player (*n* = 39). The questionnaire ad-dressing the key factors of oral-health-related topics and adjusted to the special characteristics of elite athletes was recently introduced and is described in detail in Schwesig et al. [3].

#### 2.3.2. Evaluating the Individual Oral Health Situation, Dental Examination

All included soccer players underwent a thorough dental examination by an experienced and calibrated dentist (CG). Following this dental examination widely used dental indices and scores were recorded [3,4,37,38]. The scientifically recognized DMFT (range from 0 to 28) index was used to assess the respective dental situation (number of teeth, caries, missing teeth, teeth destroyed by dental trauma, non-cariogenic lesions, restorations, restored teeth) in each examined participant [23]. The index refers to the number of teeth [12,39]. Wisdom teeth were not included. The periodontal conditions were evaluated using well established PSI index (periodontal screening index) [40]. This index provides information about the periodontal situation and periodontal treatment needs of the participants included in the study. The PSI ranges from 0 to 4. Two indices were used to assess the individual oral hygiene situation of each included soccer player. One was the papillary bleeding index (PBI) and the other was the approximal plaque index (API). Both scores are well documented and are mostly used in comparable investigations [28]. The scores range from 0 to 100% (PBI, API). The higher the score, the poorer the individual oral hygiene at home.

#### 2.3.3. Anthropometry

Body weight and height were measured using a weight scale and stadiometer. Body fat was captured using bioimpedance analysis (BC 545 digital scale, Tanita, Tokyo, Japan).

#### 2.3.4. Grip Strength

The measurement of grip strength was based on ASHT (American Society of Hand Therapists) guidelines and performed using a hand dynamometer (SH5001 SAEHAN, Changwon-si, Republic of Korea). The test procedure following the ASHT guidelines includes several important aspects [41].

A detailed description of the execution of all measurements is presented in Schwesig et al. [3]. All values from the left and right side were added (absolute combined grip strength) and related to the body mass (relative combined grip strength; kg/kg bm) based on Quinney et al. [42].

#### 2.3.5. Posturography

The Interactive Balance System (IBS, neurodata GmbH, Vienna, Austria) was used to evaluate postural stability, regulation, and weight distribution. This dynamometric assessment (sampling rate: 32 Hz) was extensively investigated concerning validity and reliability [43]. Furthermore, the IBS offers the possibility to compare all data with a pool of 1724 asymptomatic reference data [44]. The procedure of the measurement (e.g., instructions, test positions and conditions) was already described in detail in several publications of our work group.

#### 2.3.6. Endurance Test on Treadmill

The soccer players performed a treadmill test under laboratory conditions at increasing speeds to determine their running speed at specific lactate thresholds (quasar, h/p/cosmos, Traunstein, Germany). The test started at a speed level of 7.2 km/h (increment of 1.8 km/h every 3 min, slope was set at 0%) and finished at a speed level of 18.0 km/h [45]. The duration of the break (lactate measurement) between the different speeds was 45 s. Heart rates were monitored (Polar Team Pro; Polar Electro Oy, Kempele, Finland) using short-range telemetry with a 1 s recording interval, before, during and after the treadmill test. The heart rate value was recorded at the end of each running speed level. The lactate concentration was measured in hemolyzed whole blood using an enzymatic lactate analyzer (Super GL easy; Dr. Müller Gerätebau GmbH, Freital, Germany) to determine three running levels. These three levels (v2 = velocity at 2 mmol/L lactate threshold, v4 = velocity at 4 mmol/L lactate threshold, and v6 = velocity at 6 mmol/L lactate threshold) were the basis for the judgment of the general endurance performance.

#### 2.3.7. Sprint Tests

Sprint and jump tests were performed during the second and third week of the season to minimize the risk of injuries (time point of physical peak performance). Before sprinting and jumping, the soccer players performed the usual match warm-up (duration of running section: 20 min) guided by the athletic coach.

Sprint performance was assessed via photoelectric cells (AF Sport, Wesel, Germany) placed at three points: 0 m, 10 m, and 30 m on a natural grass soccer field. All soccer players started 50 cm behind the starting line from a state of complete rest. The start was independent (without any start signal from the investigator), so the reaction did not play a role. The players performed three trials and the best time was used for the statistical analyzes.

#### 2.3.8. Jump Tests

Vertical jump performance was assessed using counter movement jumps (CMJ) and squat jumps (SJ) on a jump mat (Chronojump Boscosystem, Barcelona, Spain), which provides valid and reliable measurements [46] in a training room for the players’ physical preparation. In the CMJ, players initiated the movement from an upright stance, lowered to roughly 90° knee flexion, and directly executed an explosive vertical jump with hands fixed on the hips. The SJ followed the same setup, but players maintained the squat position briefly before take-off to avoid countermovement. For both tests, correct execution required stable landings, fixed arm position, and full extension at take-off. Each soccer player completed three trials, the highest jump height was used for statistical analysis.

#### 2.3.9. Collection of Missing Days and Matches

Publicly available statistical data of an internet-related databases was used to determine the periods of injury and sickness (https://www.transfermarkt.de/; access: 15 July 2025) over a period of one to four seasons (21/22; 22/23; 23/24; 24/25). This database provides information concerning match performance (e.g., playing time, goals, assists) and medical history (e.g., type of injury or sickness, time interval, number of missing days and matches). Finally, the determined sick and injured days and missing matches were divided by the number of seasons to make them comparable.

### 2.4. Statistical Analysis

The statistical analysis was conducted using SPSS version 31.0 for Windows (IBM, Armonk, New York, NY, USA). At first, all variables were tested for normal distribution (Shapiro–Wilk Test).

The sample size calculation is based on the estimation of the correlation between variables of different dimensions (e.g., anthropometry, oral health, physical performance). Assuming a correlation of 0.65 and requiring a 95% confidence of two-sided width of 0.4 (e.g., 95% CI = (0.41; 0.81)) results in a sample size of 36 participants.

Descriptive statistics were reported depending on the different scales level (metric: mean, standard deviation (SD)); ordinal: median, percentile (interdecile range, IDR).

Associations between metric (Pearson’s correlation) or ordinal (Spearman correlation) variables were analyzed and interpreted as small (0.3), medium (0.5), and large (0.6) [47]. A r > 0.7, respectively r^2^ > 0.5 (explained variance > 50%) was defined as relevant. According to the sample size of *n* = 39, the critical value for the product moment correlation based on a two-sided *t*-test and a = 5% is r ≈ 0.310 [48].

For metric scaled parameters on both sides (dependent and independent variables), linear regression analysis (method: inclusion) was conducted. Odds ratios (ORs) including 95% CI, *p*-value, and corrected r^2^ as an estimator for the explained variance of the model were reported.

## 3. Results

### 3.1. Normal Distribution

The following parameters were not normal distributed: all dental scores, combined grip strength [kg/kg BM] in maximal intercuspal position of the mandibular, missing days, missing matches, frequency band 1 (F1), F5–6, stability indicator (ST), weight distribution index (WDI).

### 3.2. Results of Dental Examination and Performance Diagnostic

Table 2 contains a descriptive summary of the examined parameters of the several dimensions (e.g., dental health, physical performance).

The soccer players showed a high level of oral health, especially the MT was very low (0.18 ± 0.56). Compared with reference data for the postural stability, the stability indicator (20.0) moved around the percentile 75 (20.4). Consequently, the postural stability of the investigated soccer players is comparatively low compared to an asymptomatic matched (age and gender) sample. According to the anterior-posterior weight distribution, there is a noticeable reduction in pressure on the heel (42% vs. 58% forefoot). In contrast, the athletes were rather good balanced in lateral direction (left vs. right: 51 vs 49%; Table 2). Based on reference soccer data, the endurance performance can be rated as average, the jumping performance as above average and the sprinting performance, especially regarding the 10 m sprint, as below average.

### 3.3. Results of the Dental Examination and Anamnesis

Concerning several oral health statuses, the following aspects were examined using a questionnaire (*n*/%):previous illnesses (4/10%), medication intake (1/3%), number of sick days in the past two years (not sick: *n* = 3/ 8%), injured in the past two years (26/67%), any past operations (21/54%), any traumatic injuries in the mouth (7/18%), currently tooth pain (2/5%), currently bleeding gums (6/15%), currently grinding teeth (6/15%), currently tension of the temporomandibular joint (3/8%), any previously treated orthodontically (18/46%), regular check-ups (36/92%), number of teeth brushed per day (two times daily: 34/87%), additional oral hygiene procedures (21/54%), satisfaction with oral hygiene (36/92%), changes in the teeth (16/41%), negative impact on competitive sport (5/13%).

Table 3 described the association between the dental anamnesis on one side and the number of missing matches and sick and injured days on the other side.

Apart from the aspect of “undergo any orthodontic treatments”, all other dental aspects unexpectedly showed disadvantages for the soccer player without dental problems.

### 3.4. Associations Between Sick and Injured Days and Missing Match vs. Physical Performance Parameters vs. Oral Health Scores

The associations between missing days as dependent variables and the metric scaled physical performance parameters and dental scores (independent variables) are presented in Table 4. We only used independent physical performance parameters in the several categories in the model (e.g., jump: CMJ vs. SJ: r = 0.95; sprint: 10 m vs. 30 m: r = 0.92; strength: grip strength in both conditions: r = 0.88; endurance: v4 vs. v2: r = 0.93; v4 vs. v6: r = 0.98) in order to avoid interactions. Missing days and missing matches were also highly correlated with each other (r = 0.99). Therefore, we only used missing days for the analysis.

We did not find any relevant prediction power (explanation of variance) of physical performance parameters and dental scores regarding the number of missing days due to illness or injury. The explained variance ranged from 1% (sprint 30 m, metabolic recovery) to 7% (cardiac recovery).

## 4. Discussion

### 4.1. Summary of the Main Findings

The main aim of the present study was to evaluate possible associations between oral health and physical performance parameters related to the number of sick and injured days and missed matches. This is one of the few studies that examines dental health and care, several physical performance parameters and the number of sick and injured missing days and matches, respectively in professional soccer players in Germany. The special quality of this data collection was the holistic analysis of these complex dataset and the analysis of possible associations. The hypothesis that dental health has a relevant impact for the risk of sickness and injuries on one hand and the level of physical performance on the other hand, could not be confirmed.

### 4.2. Interpretation in the Context of the Literature

According to the very high level of oral health (DMFT: 3.4, PSI: 1.3; API: 27%; PBI: 18%) related to the low number of dental problems (e.g., tooth pain: 5%; traumatic injuries in the mouth: 18%; tension of the temporomandibular joint: 8%), there was no possibility to detect any relevant correlations to the missing days and matches based on diseases and injuries.

In particular, the DMFT values (3.4 ± 3.6) displayed a low caries burden. In accordance with the oral hygiene parameters (API, PBI) and the periodontal values (PSI) the reported results show that the investigated soccer players received an excellent dental care. The F-value (2.91 ± 3.09), representing the restored teeth, was not very high and accounts for more than 90% of the total DMFT value. The M-value (0.18 ± 0.56), representing missing teeth is compared to other published investigation very low [19]. Also with regards to other oral health conditions the examined soccer players showed considerably healthy conditions. Schulze et al. [12] reported untreated dental caries in 29% of adults and a DMFT of 6.7 (age range: 20–34 years) based on a comprehensive data collection from the National Institute of Dental and Craniofacial Research Health. DMFT values of elite soccer and rugby players (age range: 21–27 years) ranged from 2.7 to 5.7 [2,6,19] as a sign of a higher level of oral health. Gallagher et al. [19] investigated 352 athletes in a comparable age (mean age: 25 years; age range: 18–39 years) from eleven sports. The most commonly reported impacts were oral pain (30%) and difficulties with eating (35%). Gallagher et al. [19] described that 69% of athletes rated their oral hygiene as good/very good compared with 92% of soccer players in the present work. Comparing the oral health situation of the examined German soccer players with results focusing on representative populations in Germany, the results in the present investigation correspond to the recently published results [34,49,50].

Gay-Escoda et al. [6] examined the oral health status of professional Spanish soccer players (*n* = 30), especially in relation to the risk of sport injuries. The authors calculated no relevant correlations between Hein plaque index (PI; r = 0.42) and Ramfjord teeth probing pocket depth (PPD, r = 0.39) to intrinsic muscle injuries. These findings are in line with our results. The largest regression coefficient (0.07) was calculated for the heart rate recovery related to the number of missing days. Consequently, we had to reject the main hypothesis (oral health conditions have a relevant impact on the number of missing days) of this work.

Based on own reference data (*n* = 163; Schwesig et al. [44]) regarding postural stability and regulation, the level of postural stability is slightly below average. The stability indicator was 20.0 and moved around the percentile 75 (20.4) compared with age and gender matched data. The foot coordination (parameter: synchronization = 605) was exactly at the median of the matched sample (606). We could observe a slight unloading of the heel (41.8%; median: 45.1%). In contrast, the medio-lateral weight distribution was very well balanced (50.7%; median: 50.2%).

Haugen et al. [51] investigated the sprint performance for different distances (10 to 40 m) depending on several performance levels, positions and age groups. The authors reported 10 and 30 m times of 1.58 s and 4.04 s for 4th division players. Barrera et al. [52] described the same values for 10 m (1.57 s) for professional male soccer players from the second league in Portugal but strongly worse values for 30 m (4.55 s). Our tested soccer players showed a clearly lower level over 10 m (1.79 s) and a comparable sprinting performance over 30 m (4.17 m).

The jumping performance (CMJ: 44.5 ± 5.42 cm; SJ: 44.1 ± 5.23 cm) moved at a high level compared with several studies [51,52,53,54]. Only Bongiovanni et al. [53] collected higher values for CMJ differentiated by high (52.1 ± 5.1 cm) and low performers (47.9 ± 4.6 cm). However, the authors investigated first league soccer players (*n* = 21; age: 27.2 ± 5.1 years) from Italy. According to the SJ performance, Asimakidis et al. [55] reported within a comprehensive systematic review a range from 29.8 to 52.8 cm (median: 38.0 cm) across all age groups and performance levels. Compared with these values, the jumping performance of our sample of soccer players was above average (44.1 ± 5.23 cm). Barrera et al. [52] measured a lower value (36.0 cm) and a larger difference (1.2 cm vs. 0.4 cm) compared to the CMJ (37.2 cm). The almost identical jump heights (44.5 vs. 44.1 cm) of our sample indicate impaired stretch-shortening cycle mechanics, particularly of the slow stretch-shortening cycle during the CMJ [56].

The endurance performance measured on treadmill under standardized conditions (v2: 11.9 km/h; v4: 14.9 km/h; 16.4 km/h) was around the median (12.2 km/h, 15.0 km/h, 16.5 km/h) based on the reference data by Schwesig et al. [45] at all levels (v2, v4, v6). Basically, the evaluation of endurance performance in soccer is underrepresented in the scientific literature, even though this test is conducted at least once per season as standard practice. For example, the comprehensive and up-to-date systematic review by Asimakidis et al. [55] surprisingly did not contain any data in this regard.

### 4.3. Limitations

The data collection has some relevant limitations. Firstly, the most important limitation is the lack of scientific investigations with adult professional soccer players in the field of performance diagnostic and dental sports medicine, especially the connection of both aspects in detail. Based on our long-term experiences in this field, it is very difficult to motivate the responsible supervisors of the team (manager, coach, team doctor) to engage in such complex and longitudinal data collection. Against this background, we had to use a non-scientific source (transfermarkt.de) to obtain information about injury days and missed matches. This means that the data on absences may be underestimated or inconsistent (e.g., not every injury is officially recorded).

Secondly, future studies should focus more on the aspects of performance (different leagues) and age (junior vs. adults). In this context it would be meaningful to investigate different teams in order to judge different strategies of dental care and physical performance diagnostic.

Thirdly, important confounder factors (e.g., quality of sleep, nutrition, diet logs, and different training loads) should be considered in future study designs. For example, an interesting question could be to investigate the influence of supplements consumption, especially with a high percentage of sugar and carbohydrates, regarding the improvement of physical performance and the possible influence on oral health simultaneously.

Finally, the different appointments of examination (e.g., endurance diagnostic, posturography and grip strength testing vs. sprint and jump tests) are a critical point from a methodological point of view and might influence the presented results.

## 5. Conclusions and Practical Implications

This project provides a valuable, comprehensive dataset concerning common and dental health status, common and soccer specific physical performance evaluations in several dimensions (e.g., endurance, strength, speed) for German elite soccer players. In connection with the collection of data on days lost due to illness and injury and the resulting number of missed matches over three seasons, this complex and comprehensive approach is one strength of this study.

These primary outcomes, which were highly correlated with each other seems not to be associated with the predictors mentioned above. Presumably, this unexpected main result is probably caused by the excellent dental health status of the soccer players. In order to enhance the scope of validity, a more heterogeneous sample (e.g., several performance levels and age groups) should be taken into consideration.

However, according to our examinations and the correlations from recently published data, adjusted screening protocols, like biannual dental checks (e.g., tooth vitality, traumatic injuries, cariological, periodontal and restorative screening) should also be part of a sufficient medical health (e.g., orthopedic and cardiological) and physical performance (e.g., posture, strength, endurance, speed) check.

In order to avoid absences or limitations due to tooth inflammation or pain, especially in professional soccer, regular check-ups and preventive measures as part of dental visits seem advisable and should be implemented.

## Figures and Tables

**Figure 1 sports-13-00417-f001:**
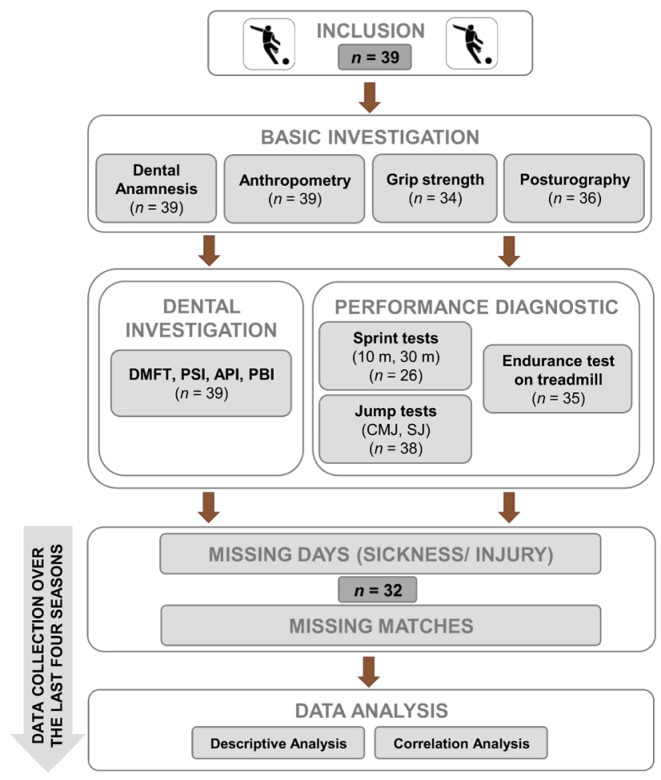
Flow diagram regarding content of entire project. All investigations were included only once in data analysis, whereas days lost to sickness or injury and number of missed matches were collected over a period of four years and then normalized to one year. Using this approach, distortion effects were avoided (e.g., serious injury in one year).

**Table 1 sports-13-00417-t001:** Demographic and anthropometric characteristics of entire sample (*n* = 39). SD = standard deviation.

	*n*	Mean ± SD	Range
Age [years]	39	24.6 ± 4.23	17.1–34.4
Height [m]		1.84 ± 0.06	1.71–1.93
Body mass [kg]		81.1 ± 7.65	67.9–96.2
BMI [kg/m^2^]		24.0 ± 1.73	20.4–28.0
Fat [%]		13.1 ± 3.37	6.7–20.4
Combined grip strength [kg/kg BM] in physiological rest position of the mandibular	39	1.26 ± 0.15	0.77–1.52
Combined grip strength [kg/kg BM] in maximal intercuspal position of the mandibular	35	1.31 ± 0.16	0.81–1.54
Missing days per season [d]	32	56 ± 66	0–360
Missing matches per season		7 ± 7	0–41

**Table 2 sports-13-00417-t002:** Results of dental screening and postural performance. Heel = percentage of weight distribution forefoot vs. hindfoot with reporting of heel loading; Left = percentage of weight distribution left vs. right with reporting of left side loading; Synchronization = foot coordination.

	Mean ± SD	95% CI
**Dental Health Scores (*n* = 39)**		
DMFT	3.41 ± 3.60	2.24–4.58
DT	0.31 ± 0.66	0.10–0.52
MT	0.18 ± 0.56	0.00–0.36
FT	2.92 ± 3.09	1.92–3.92
PSI	1.28 ± 0.76	1.04–1.53
API [%]	27.2 ± 16.0	22.0–32.4
PBI [%]	18.2 ± 15.6	13.1–23.2
**Posturography (*n* = 36)**		
F1	19.0 ± 5.62	17.1–20.9
F2–4	8.87 ± 1.84	8.25–9.49
F5–6	3.54 ± 0.75	3.29–3.80
F7–8	0.70 ± 0.16	0.65–0.75
ST	20.0 ± 4.55	18.5–21.6
WDI	6.39 ± 2.60	5.51–7.27
Heel [%]	41.8 ± 8.67	38.9–44.8
Left [%]	50.7 ± 2.59	49.8–51.6
Synchronization	605 ± 136	559–651
**Sprint Performance [s] (*n* = 26)**		
10 m	1.79 ± 0.09	1.75–1.83
30 m	4.17 ± 0.15	4.10–4.22
**Jump Performance [cm] (*n* = 38)**		
CMJ	44.5 ± 5.42	42.7–46.3
SJ	44.1 ± 5.23	42.4–45.8
**Endurance Performance [km/h] (*n* = 35)**		
v2	11.9 ± 1.43	11.4–12.4
v4	14.9 ± 1.11	14.5–15.3
v6	16.4 ± 1.00	16.0–16.7

**Table 3 sports-13-00417-t003:** Number of missing matches and sick and injured days in relation to selected aspects of dental anamnesis.

	Number of (Mean ± SD)			
Aspects of the Dental Anamnesis	Missing Matches		Sick and Injured Days	
	Yes	No	Yes	No
Do you have toothache?	*n* = 1: 3 ± 0	*n* = 32: 7 ± 7	27 ± 0	55 ± 66
Did you have gum bleeding in the past?	*n* = 4: 3 ± 4	*n* = 29: 7 ± 8	27 ± 38	58 ± 67
Do you suffer from tensions around you jaw joint/shoulder/neck?	*n* = 2: 5 ± 4	*n* = 31: 7 ± 8	31 ± 18	56 ± 67
Did you undergo any orthodontic treatments?	*n* = 13: 9 ± 10	*n* = 20: 5 ± 4	76 ± 92	40 ± 35
Do you regularly go to the dentist?	*n* = 31: 7 ± 7	*n* = 2: 2 ± 2	57 ± 66	12 ± 16
Do you use any additional oral care products?	*n* = 17: 9 ± 9	*n* = 16: 5 ± 5	68 ± 80	40 ± 43
Are you happy with your oral health?	*n* = 30: 7 ± 8	*n* = 3: 5 ± 2	57 ± 68	33 ± 9

**Table 4 sports-13-00417-t004:** Linear regression for missing days (dependent variables) and physical performance parameters and oral scores as independent parameters. BM = body mass.

Independent Variables	Statistical Values of the Model			
	r_corr_^2^	OR	95% CI	*p*
**Physical Performance Parameters**				
CMJ [cm]	0.04	3.19	−1.04; 7.42	0.134
Sprint 30 m [s]	0.01	93.6	−80.7; 268	0.279
v4 [km/h]	0.05	−16.8	−38.4; 4.83	0.123
Lactate degradation rate per minute [mmol/L/min]	0.01	−57.5	−160; 45.2	0.261
Recovery heart rate (relative) [%/min]	0.07	−22.9	−49.6; 3.85	0.091
Relative combined grip strength in rest position [kg/kg BM]	−0.03	30.8	−124; 186	0.688
F1	−0.04	−0.30	−4.77; 4.17	0.892
F2–4	−0.04	1.08	−12.6; 14.7	0.873
F5–6	−0.02	−9.94	−43.3; 23.7	0.546
F7–8	−0.02	−47.4	−205; 111	0.544
ST	−0.02	−1.66	−7.15; 3.83	0.540
WDI	−0.04	0.08	−9.58; 9.73	0.987
Synchronization	−0.02	0.06	−0.12; 0.25	0.487
**Dental Scores**				
DMFT	−0.03	0.85	−5.74; 7.44	0.794
API	−0.03	0.24	−1.24; 1.72	0.743
PSI	−0.03	−2.53	−33.8; 28.8	0.870
PBI	−0.03	−0.22	−1.74; 1.31	0.774

## Data Availability

The data are not publicly available due to ethical restrictions. The anonymized data are available upon request from the first or corresponding author.

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
