# Peer review of "The Impact of Oral Health on the Number of Missing Matches and Physical Performance in Elite Male Soccer Players"

_sports, 2025, doi:10.3390/sports13120417_

Round 1

Reviewer 1 Report

Comments and Suggestions for Authors

Manuscript: “The Impact of Oral Health on the Number of Missing Matches and Physical Performance in Elite Male Soccer Players”

General comment

The manuscript argues an interesting topic for sport performance. Precisely, the role of oral health on physical performance and success in matches.

Although the manuscript is well written, authors should clarify some methodological issues.

Specific comments

Abstract

-Line 25 “Thirty-nine male athletes … were examined concerning several dental parameters (DMFT, PSI, API, PBI)…”. What is the meaning of these acronyms? This is the first time they appear in the abstract.

-Line 30 “We couldn`t find any relevant (r > 0.7)…”. Please also report the level of significance.

-Line 32 “…high level of oral health (MT: 0.18 ± 0.56)… The endurance capacity (v4: 14.9 33 ± 1.11 km/h)…”. What is the meaning of MT, v4?

Introduction

-Line 46 “In high-performance environments, elite soccer players are expected to maintain peak physical, mental, and physiological conditions.” Since no references support this statement, I suggest considering the following: “Pillitteri G, et al. Relationship between external and internal load indicators and injury using machine learning in professional soccer: a systematic review and meta-analysis. Res Sports Med. 2024;32(6):902-938. doi:10.1080/15438627.2023.2297190”

-Line 114 “Our primary goal was to evaluate the oral health conditions and to inform and prepare targeted interventions that embed oral health care within elite soccer routines.” I agree with the first part of the sentence, that is, that the primary goal was to evaluate the oral health conditions in soccer players, but is it with this study that the authors intend to prepare targeted interventions that embed oral health care within elite soccer routines?

- What is the hypothesis of the study?

Methods

-Line 121 “How do the authors determine that a sufficient sample size has been recruited?” Please provide more details on sample size power analysis in the statistical analysis section. Was it conducted a priori?

-Authors failed to report recruitment process.

- The procedure should be better described. If I understand correctly, the authors collected data on: 1) “Medical and dental anamnesis and clinical examination”; 2) “Basic physical diagnostic”; 3) “Specific performance diagnostic”. And they matched information on the number of sick and injury days and missing matches using the website from https://www.transfermarkt.de/. If so, the procedure should be rewritten with more scientific rigor. Also specify when the data were collected. Furthermore, state who performed the assessments and tests. Were they always the same researchers? State the data collection setting. Where were the measurements taken? Furthermore, was the time-of-day effects taken into account? It is widely known that this can impact performance results.

-Line 130: As above, “Apart from the sprint and jump tests, all these examinations were conducted at the beginning of the preparation phase of each season (last two weeks in June).” This is a critical point, and it should be reported among the limitations of the study that the time points are different. When were the sprint and jump tests performed?

-Line 156: The oral health status was evaluated using a questionnaire. Is this questionnaire standardized and validated?

- Procedures for some measurements are not reported and authors refer to previous studies (e.g. for Medical and Dental History, Grip strength, Posturography). Authors should exactly report what they did, describing the procedure in detail so that the study can be replicated.

-For posturography, why did they set a sampling rate of 32 Hz? Please report on the rationale, supported by the literature, for this choice.

- Sprint Tests and Jump Tests: as above, more details on setting and procedure should be reported. For example, where were these tests conducted? Was there a warm-up?

Discussion

-Authors refer to a hypothesis that is not clearly reported in the introduction section.

-Please report specific practical implications of the study emphasizing the applications in soccer.

-Authors failed to report the strengths of the study.

Author Response

Dear Reviewer 1 thank you very much for your review and all the helpful comments and recommendations. 

Please see the attachment including the point-by-point response. Thank you.

Reviewer 2 Report

Comments and Suggestions for Authors

The introduction generally provides a good research background; however, I have a few minor comments:

  • In my opinion, it should be supplemented with more recent studies (2023–2025) concerning inflammatory biomarkers and athletic performance (e.g., IL-1β, CRP in the context of the oral microbiome).
  • The differences between various sports disciplines should be highlighted, as findings from football may not be fully transferable to other team sports.
  • The excessive use of value-laden expressions such as “alarmingly poor oral health” or “considerably high impact” should be avoided. In scientific writing, precise and data-based wording is preferred, e.g. “a high prevalence of oral pathology was observed” or “self-reported impairment in performance exceeded 7%.”
  • L79 – I would expand the information regarding bruxism and increased masticatory muscle activity in athletes related to stress and physical exertion. This may cause both daytime and nocturnal bruxism, which in turn can contribute to TMD development. Regarding the impact of bruxism on oral health in athletes, the following reference may be useful: DOI: [10.1055/a-2588-0766]. For the relationship between bruxism and TMD, and epidemiological data, please see DOI: [10.17219/dmp/201376].
  • Please add a clear research hypothesis at the end of the Introduction.

Materials and Methods

This section requires further elaboration and refinement:

  • A clear justification for selecting Transfermarkt.de as the source of injury data is missing. It is a secondary source without clinical validation — this limitation should be explicitly discussed or supported by literature showing its previous use in research.
  • It should be noted that absence data might be underestimated or inconsistent (e.g. not every injury results in an official record).
  • The number of participants (n = 39) is described as a convenience sample — therefore, the analyses should be presented as exploratory, not inferential.
  • Consider conducting a power analysis – even retrospectively – to justify the adequacy of the sample size. The authors could either use software such as GPower or refer to the following studies: DOI: [10.3352/jeehp.2021.18.17] or DOI: [10.1016/j.apmr.2025.05.013], where it is noted that a group of 34 participants is sufficient to detect large effect sizes at 90% power.
  • The phrase “over four seasons” lacks clarification on whether clinical and diagnostic assessments were repeated annually (suggesting a longitudinal design), or whether a single baseline assessment was conducted followed by a four-year observation of sports outcomes.
  • If repeated measurements were performed, please specify this clearly (e.g. “annual reassessments”); if not, the study should be described as a retrospective cohort rather than a longitudinal study.
  • Section 2.4 Statistical Analysis – I appreciate that the authors applied effect size measures; however, the interpretation thresholds were incorrectly used. Cohen (1988) emphasised that his benchmarks apply only in the absence of discipline-specific standards. In this case, the most comparable field is physiotherapy/rehabilitation, where the following effect size conventions are recommended: For individual differences (Pearson’s r): small = 0.3, medium = 0.5, large = 0.6 ;For group differences (Cohen’s d or Hedges’ g): small = 0.1, medium = 0.4, large = 0.8. Please revise the description accordingly and refer to DOI: [10.1016/j.apmr.2025.05.013].

Results

In my opinion, the results are well conducted and clearly presented.

Discussion

At the beginning of the Discussion, I recommend reorganising the structure to follow the classical format:
a concise summary of the main findings → their interpretation in the context of the literature → possible explanations for observed differences → limitations → practical implications → conclusions.
Currently, results, interpretations, and limitations are mixed together, which reduces readability.

Conclusions

In my view, the conclusions are too general and lack clear clinical implications – please revise accordingly.

Author Response

Dear Reviewer 2

Thank you very much for your review and all the helpful comments and suggestions. 

Please see the attachment including the point-by-point response. Thank you,
